# OpenReview forum: "FB-Bench: A Fine-Grained Multi-Task Benchmark for Evaluating LLMs' Responsiveness to Human Feedback"
_ICLR.cc/2025/Conference — ICLR 2025 Conference Withdrawn Submission_

### Official Review · Reviewer_ukzG · 2024-10-26

**Soundness:** 2
**Presentation:** 3
**Contribution:** 2
**Rating:** 5
**Confidence:** 4

**Summary:**

Human interactions with LLMs are often collaborative and multi-turn in nature. However, existing benchmarks cannot effectively measure the responsiveness of LLMs to human feedback. To solve this research gap, they introduce a new benchmark, aiming to measure LLMs’ responsiveness to various types of fine-grained human feedback, such as error correction.

Authors first collect a set of tasks from various domains (code / math / reasoning) and tasks tasks (extraction, translation, correction, generation, question answering). Then they check model responses how they respond to issues such as poor instruction following, logical errors, factual errors, incomplete answer and lacking in response cohesiveness. Authors also introduce an adversarial case where response are correct to see if models will maintain answers when challenged by users.

**Strengths:**

1.	This topic is important for better Human-AI collaboration and has not been well-studied previously.
2.	The paper is well-written and supported by clear illustrations that make it easy to follow.
3.	The observation that models’ capability to do error correction is distinct from their ability to do response maintenance (in Figure 5) is interesting and insightful. I’m surprised by the observation that there is a large gap between models doing error correction (generally good) and their ability to do response maintenance (generally less good), especially “pronounced among LLMs developed outside of China” (line 379).

**Weaknesses:**

1.	I might be missing something but currently, it’s not stated whether the data contains only the prompts or also the first turn assistant responses (which should ground the feedback). The supplementary material suggests that the first turn assistant responses are fixed but I’m not sure if it’s just for that sample or more generally the case. If it is, which model generated that first turn assistant responses? Furthermore, the data in the supplementary materials suggests that many of the data samples are in Chinese but this was not mentioned in the paper – if there are different languages, please clearly state it in the paper and report the proportion of data in each language. It’s extremely relevant in grounding the understanding of model performance (e.g. models trained by providers based in China might be better in tasks if they contain Chinese). Figure 10 and 11 show identical examples in both Chinese and English – are models tested on both languages?
2.	In line 272, it’s stated that GPT-4 is used to identify issues with responses and provide with annotators refining the pre-annotated elements. The paper fails to state the proportion of cases that the human annotators agree with GPT-4. Ideally, there will be some comparison between GPT4-human agreement and human-human agreement to better understand the reliability of this data curation pipeline. Can the authors also clarify if each sample is reviewed by more than one reviewer? What are the background of these reviewers (e.g. Amazon MTurks?) and what are the guidance given to them to do this task? Currently, it’s missing a lot of critical information as an evaluation dataset contribution.
3.	In line 277, it’s not clear why the authors chose those three models as difficulty filters as they all seem of similar sizes. Also, it will be better to give a sense of the proportion that scores 1 by each model, to get a sense of the filtering rate.
4.	In both the results and analysis sections (3.2 and 3.3), there are no attempt to relate the empirical results to possible explanation identified by other studies, which makes reading the results feel very non-grounded and mostly based on the authors’ guesses/hypotheses. One possible way to relate this to literature is by connecting the low response maintenance rates to the concept of model sycophancy proposed by [1] Another way to relate this is possibly from the cultural lens of what types of behavior are more encouraged in models trained within and outside of China [2]

[1] https://arxiv.org/abs/2310.13548
[2] https://arxiv.org/abs/2410.02677

**Questions:**

NA

**Details Of Ethics Concerns:**

The paper contains a sentence on "Finally, the annotators act as the reviewers to refine all pre-annotated elements of each sample." (line 275) but it was not mentioned anywhere who the annotators are and if they were recruited and compensated adequately.

---

### Official Review · Reviewer_2zLt · 2024-10-27

**Soundness:** 2
**Presentation:** 2
**Contribution:** 3
**Rating:** 3
**Confidence:** 4

**Summary:**

This paper introduces a benchmark on evaluating LLMs’ responses to human feedback. They collect data over multiple task domains and introduce feedback that either critiques an LLM’s response. The LLM should then respond with either a correction or maintain its response, in cases when the feedback is not justified.
Using their 734 benchmark samples, they evaluate a broad spectrum of current LLMs, using GPT-as-a-judge with a checklist. The paper provides an analysis of the evaluated LLMs’ strengths and weaknesses with respect to feedback.

**Strengths:**

The main strengths of this paper lie in the originality of the benchmark: To the best of my knowledge, no similar benchmark exists. Additionally, it covers a gap in current LLM research, which is the analysis of LLMs’ ability to engage in meaningful multi-turn conversations, including responding to feedback. I also believe that it is interesting that the paper analyzes how some LLMs are overly responsive to feedback, and do not stay consistent with their answer, even when the feedback is wrong. I also commend the use of human annotators in constructing the benchmark (although I have some comments about that in the weaknesses section).

**Weaknesses:**

There are two main weaknesses (besides my smaller questions in the field below) that I would like to highlight.

1. The annotation setup does not seem very rigorous: The paper fails to mention who the human annotators are, how many there are, how much they were paid etc. Additionally, it does not report any IAA or any quality assessment of the benchmark itself (i.e. how can we be sure to be able to trust the quality of the benchmark?) I would have also liked to see a discussion on why a human-LLM set-up was needed and why they have decided not to use a completely human written benchmark? (especially considering the small size of the benchmark - 734 samples)

2. The writing lacks clarity: It is not clear to me what the evaluation setup exactly looks like. Is the input a prompt - response - critique and the output a response to the critique? If yes, does it matter that the first response has potentially not been generated by the same model that will generate the second response? I further would have liked to see an analysis of the “fine-grained” feedback: what makes it fine-grained? (more questions regarding clarity of writing below). The paper also does not mention the specific data sources (i.e. citing the specific datasets).

**Questions:**

- Is the feedback human written?
- 2.2.2 Is that before or after the user’s feedback?
- Line 204: is unprofessional the right name?
- 2.3.2.: The scoring of the model response is done based on a checklist? Which? Would have been good to point to the specific checklist.
- 259: human preference data: where does it come from? No citation…
- 267: Who are the annotators? What is the agreement?
- Figure 3 does not show for me at all.
- It would be great to see how fine-grained the feedback actually is and how diverse it is.

---

### Official Review · Reviewer_bJeE · 2024-10-30

**Soundness:** 3
**Presentation:** 3
**Contribution:** 2
**Rating:** 3
**Confidence:** 5

**Summary:**

The paper introduced a new Chinese multi-turn benchmark to evaluate LLMs appropriate responsiveness to user’s feedback including the ability to correctly fix errors or maintain its correct response when challenged by the user.
The FB-Bench is built based on a taxonomy covering 8 different task types, 5 different error types and 9 feedback types.

The author evaluated several different open and closed-sourced models and presented some insights on how current models perform on this task. They used LLM-as-judge to evaluate the responsiveness of LLMs based on a predefined checklist.

While I appreciate the author's effort in making this benchmark, I am not super impressed by this work as a publication in a top-tier ML conference.

**Strengths:**

- This is a useful resource specially for as non-English benchmark further advancing models development for language other than English
- Extensive evaluation of several models, with some interesting findings

**Weaknesses:**

- One major issue I had while reading the paper was that I only realized this is a Chinese resource more than halfway through the paper by going to Appendix. I strongly suggest the author specify the language of your resource early on.
- Some Phrases could be tone-downed a bit, .e.g, three-tier hierarchical taxonomy. For one, I would call this an evaluation framework and not a taxonomy, and I am not sure which aspect of it makes it hierarchical.
- It would have been nice to extend this to a multilingual benchmark covering other languages

**Questions:**

1- What future research directions do the authors anticipate their work will enable?

2- Mention language of your benchmark in the abstract/introduction

3- Line 52: dialugues -> dialogues

4- Some correspondence between Figure 2 and its caption would be nice. The reader benefits from understanding what 2 deficiency types or 9 feedback types refer to in the figure? Is this 2 deficiency or 5?

5- missing citation. There are some previous works introducing the concept of checklist for LLM-as-judge style evaluation and similar post-filterng strategies. It’s nice to acknowledge those work:
Lin, Bill Yuchen, et al. "WILDBENCH: Benchmarking LLMs with Challenging Tasks from Real Users in the Wild." arXiv preprint arXiv:2406.04770 (2024).
5- providing examples in Table 1 would help the reader to better understand the  user feedback

6- It is not clear how the checklist weights are determined? Seems a bit arbitrarily?

7- Are there examples of the checklist? Are they defined at instance-level or task-level?

8- Line 377: similar findings repeated twice!

9- Line 379, caption: I guess what the author is trying to convey is that LLMs that are mostly optimized for non-chinese language, are performing worse on FB-Bench? Also worth mentioning in footnote which models are developed in China

---

### Official Review · Reviewer_TSVp · 2024-11-04

**Soundness:** 2
**Presentation:** 4
**Contribution:** 3
**Rating:** 5
**Confidence:** 4

**Summary:**

This paper introduces a new benchmark, FB-Bench, designed to evaluate the responsiveness of LLMs to human feedback in real-world scenarios. The authors argue that existing benchmarks primarily focus on single-turn dialogues or multi-turn dialogues with independent user inputs, which fail to capture the complexity of human feedback in iterative interactions. FB-Bench addresses this gap by offering a more nuanced evaluation framework. FB-Bench is the first systematic benchmark focused on evaluating LLMs' responsiveness to human feedback across diverse real-world tasks. It includes 734 curated samples covering eight task types, five types of response deficiencies, and nine types of user feedback. The authors develop a hierarchical taxonomy that organizes human-LLM interactions into three components: user queries, model responses, and user feedback. This taxonomy supports two main interaction scenarios: error correction (where the model must improve its response based on user feedback) and response maintenance (where the model must maintain a correct response despite potentially misleading feedback). The evaluation is conducted using a weighted checklist specific to each sample, with GPT-4o (August) acting as a judge to score follow-up responses based on human-curated reference responses. The paper evaluates 31 popular LLMs using FB-Bench and reveals significant performance discrepancies between error correction and response maintenance scenarios. Notably, most LLMs perform better in error correction than in maintaining correct responses when faced with misleading or irrelevant feedback.

**Strengths:**

Originality:
While multi-turn evaluation has been attempted before, the authors present a new perspective focusing on error correction and responses maintenance, which seem less explored.
The authors use a checklist when doing LLM judge evaluations for more finegrain judgments.
Clarity:
The paper is well organized and easy to read, with nice structure and clean figures.
Significance:
The authors find an interesting comparison between error correction and responses maintenance (which inherently seem a bit in tension). The authors find most models have superior error correction performance, suggesting a current weakness in LLMs.
The benchmark itself fills an existing hole for evaluating models on multi-turn conversations.

**Weaknesses:**

Out of distribution first turn:
The first turn of the data is collected from some less human preferred response rather than the model tested itself. Naturally, this means the first turn is already out of distribution for the model. It is unclear if this invalidates the real-world applicability of these results. For example, the paper finds Claude-3.5-Sonnet has very low response maintenance (relatively). Would this still happen if Claude-3.5-Sonnet was asked to maintain its own answer? Take another example, Phi-3-small-8k-instruct has very high response maintenance, but low error correction. This would be desired behavior if the model’s own responses were usually correct. However, if the model's response is not fully correct (which is more likely because the model is very small) this response maintenance is actually a negative when deployed in the real world.

Moreover, just with respect to the pure out of distribution nature of the first turn (you are basically putting someone else’s words in the LLM’s mouth), it may be helpful to see an experiment that is single turn. In this case, the model being tested is given the original user query, and the first response in which the model is asked to correct/maintain the response. This way, no out of distribution tokens are inserted into the assistant section of the sequence. If results in this experiment are similar, it would be sufficient to show that the out of distribution first turn is, at least mechanically, not an issue.


LLM Use in Data Curation:
In lines 271-272 the authors state that GPT-4 was used to find the cause of dissatisfaction when the user’s expectations were not met. However, this feels in contradiction to the conclusion of the paper that LLMs are weak for response maintenance. That is, the human’s expectations could have been wrong or unreasonable, in which the GPT-4 annotator may hallucinate some incorrect cause of dissatisfaction via the same sycophant behavior this paper finds almost every LLM has. Annotation process is not explained in enough detail to convincingly rule out this possibility.


Analysis:
While the results for models are interesting, the analysis focuses on superfluous information rather than deeper insights. For example, the authors write on line 379 that “This disparity [between error correction and response maintenance] is more pronounced among LLMs developed outside of China”. This claim seems a bit out of place, and is not supported with concrete numbers (especially because the benchmark seems to be in Chinese, which will obviously yield better results on models trained for this language, and is a huge confounding variable). Moreover, analysis like this is not particularly tied to the core exploration of error correction vs response maintenance. As a reader, error correction vs maintenance feels more analogous to precision vs recall. Poor performance overall (like Claude-3.5-Sonnet) seems like a calibration issue, not an inherent flaw of the model. For example, any model could achieve 100% success on response maintenance with a system prompt that tells it NEVER to change its initial response. It seems error correction and response maintenance are something that could be calibrated. Looking at Figure 4, it seems “evenly calibrated” models will lie approximately on y=x. However, this assumes that error correction and response maintenance occur at the same rate in the real world, which may not be true. For example, the “smarter” a model gets, the more it should stick to response maintenance rather than error correction, since it can assume it is more skilled with respect to the human user giving feedback. This point connects back to my original critique with out of distribution first turns covered above— we can never know the real aggregate performance of a model without first understanding how often the model will get the first turn incorrect or unsatisfactory. For real-world applicability, the authors should at least consider the tension between error correction and response maintenance in their core analysis. For example, what happens when we use a system prompt that always forces error correction? What about the opposite? Will these get top accuracy in each category if used in conjunction with a suitably strong model?


LLM Judge:
While the usage of the checklist judge is interesting, the authors do not verify the judge's accuracy. This on top of the fact, the checklist itself is GPT-4 generated and seemingly unverified. For example, does the ranking change if we simply change the judge to a different strong model? Why was GPT-4o-08-06 selected as a judge? The authors should have some level of verification for judge quality and simultaneously use this to justify their judge choice.

**Questions:**

Most big questions ended up in the weakness section, however some desired clarifications are shown below:

On lines 270-275, the authors state that GPT-4 was used for annotation. Which GPT-4 is this (0314, 0613, or some turbo version)? Why was GPT-4 used instead of new models which seem to perform better?

On line 473, the authors criticize that data in previous multi-turn benchmarks are synthesized by LLMs. However, it appears that LLMs are used to synthesize a significant portion of FB-Bench’s data (lines 272-275). Why does this same criticism not apply to FB-Bench?

Figure 6 is interesting, but the very light colors on the white background are a bit hard to see. Perhaps slightly darker colors, less line opacity, and points on the axis would make it slightly more readable.

Multiple times in the paper the authors reference “advanced” models. It is a bit ambiguous what “advanced” encompasses. A more explicit definition would be helpful.

In section 2.4, the authors explain their data is from “an online chat service and human preference data”. Could the author please give more detailed information on what these exactly are, why they are reputable, and give examples from the data sources?

In line 261, the authors say they use a heuristic to identify target data, but never specify in any detail what this heuristic is, or why it works.

In line 273, the authors say GPT-4 is used to generate the judgment checklist. How are the weights for the checklist decided? How is the checklist verified? Does removing the checklist significantly change final ranking scores?

---

### Note · Authors · 2024-11-26

I have read and agree with the venue's withdrawal policy on behalf of myself and my co-authors.